# Antibacterial Effect of Endodontic Disinfections on Enterococcus Faecalis in Dental Root Canals—An In-Vitro Model Study

**DOI:** 10.3390/ma14092427

**Published:** 2021-05-07

**Authors:** Stefan Kranz, André Guellmar, Franziska Braeutigam, Silke Tonndorf-Martini, Markus Heyder, Markus Reise, Bernd Sigusch

**Affiliations:** Department of Conservative Dentistry and Periodontology, Jena University Hospital, Friedrich-Schiller-University, An der Alten Post 4, 07743 Jena, Germany; andre.guellmar@med.uni-jena.de (A.G.); Franziska_braeutigam@web.de (F.B.); Silke.Tonndorf-Martini@med.uni-jena.de (S.T.-M.); Markus.Heyder@med.uni-jena.de (M.H.); Markus.Reise@med.uni-jena.de (M.R.); Bernd.W.Sigusch@med.uni-jena.de (B.S.)

**Keywords:** chloramine-T, *Enterococcus faecalis*, temporary dressing, calcium hydroxide, sodium hypochlorite, intracanal disinfection, root canal treatment, endodontic treatment

## Abstract

*Enterococcus faecalis (E. faecalis)* is rather unsusceptible to many root canal disinfections which often cause a therapeutic problem. Therefore, the present in vitro study observed the efficiency of different endodontic antiseptics in their capability to suppress *E. faecalis*, especially inside dentinal tubules. Prior to any testing, root canals of extracted third human molars were inoculated with *E. faecalis* for 48 h. Antiseptic dressings with chloramine-T or calcium hydroxide (CaOH) for 24 h or irrigations with 1.3% sodium hypochlorite (NaOCl) were applied with n = 10 in each group. As control irrigation with normal saline was used. All treated canals were manually enlarged from size ISO 50 to 110 and the ablated dentin debris was subjected to microbial culture analysis. Bacterial colonization of the dentinal tubules up to 300 µm was verified by scanning electron microscopy and histological sample preparation. Application of crystalline chloramine-T caused total bacterial suppression inside the dentinal tubules. Dressings with CaOH showed only minor effects. Irrigation with NaOCl caused total eradication of bacteria adhering to the root canal walls, but also failed to completely suppress *E. faecalis* inside the dentinal tubules. The study showed that chloramine-T is of strong antiseptic activity and also efficient in suppressing *E. faecalis* inside dentinal tubules.

## 1. Introduction

Chemo-mechanical root canal preparation involves the removal of pathogens by mechanical shaping of the root canal walls accompanied by an alternating irrigation with disinfection fluids. The treatment of infected dental root canals by high quality disinfection is crucial to ensure healing of the apical region and long-term success of an endodontic therapy. In this regard, pathogens that have survived inside dentinal tubules or in inaccessible areas of the root canal system might cause re-infection and treatment failure [1,2].

In those cases, it was shown that in post-treatment endodontic flare-ups besides Streptococci, Lactobacilli, Actinomyces and fungi, the Gram-positive bacterial species *Enterococcus faecalis* (*E. faecalis*) is often detected [3,4]. Investigations have shown that in cases of secondary root canal infections the prevalence of *E. faecalis* can reach values up to 70% [5,6,7,8].

*E. faecalis* is also known to be rather unsusceptible towards many disinfectants commonly applied during antiseptic root canal treatment which include chlorhexidine [9,10], sodium hypochlorite [11,12] and calcium hydroxide [13,14]. Even exposure to clindamycin, tetracycline, erythromycin or treatment with Lugol’s iodine solution as well as mineral trioxide aggregate was not efficient in suppressing the species completely [15,16,17].

Unlike others, *E. faecalis* is also able to penetrate far into the dentinal tubules which ensures protection from antiseptic measures, too [2].

In order to overcome the lack of inefficient intracanal disinfection, new chemical compounds and methods are constantly under investigation for their suitability in endodontic antiseptic treatment [18,19,20,21,22,23].

Surprisingly, the already known disinfectant N-chloro tosylamide sodium salt (chloramine-T) shows promising characteristics [24]. Chloramine-T is of high chlorination and oxidation power and proofs sufficient antimicrobial behavior towards a wide variety of microorganisms including bacteria, fungi, yeasts and also human pathogenic viruses such as coxsackie, parainfluenza, adeno-, polio- and corona-viruses [25,26,27].

In contrast to sodium hypochlorite (NaOCl) which is still considered the “gold standard” in chemo-mechanical root canal preparation, chloramine-T is also of less cytotoxicity [24,27,28].

Although chloramine-T is a well-known disinfectant, studies investigating the bactericidal effect on endodontic pathogens are still rather rare. Especially for pathogens such as *E. faecalis* which are able to colonize dentinal tubules, application of chloramine-T by means of a temporary dressing might be of potential interest.

Therefore, the present in vitro study was aimed on investigating the antibacterial effect of chloramine-T, CaOH and NaOCl in their capability to suppress *E. faecalis* inside the dental root canal system and especially inside dentinal tubules.

## 2. Materials and Methods

### 2.1. Cultivation of Enterococcus Faecalis

In the present in vitro study the Gram-positive bacterial species *Enterococcus faecalis* (ATCC 29212) was used. The strain was grown under anaerobic standard conditions (80% N_2_, 10% CO_2_, and 10% H_2_) in 10 mL Schaedler fluid media (Oxoid Ltd., Hampshire, UK) for 3 h. Subsequently, bacteria were harvested by centrifugation (4000 rpm, 8 min) and washed twice with PBS. For inoculation, bacterial solutions were arranged by re-suspending the bacterial pellet in Schaedler fluid media (OD546nm of 0.5 which corresponds to 10^8^ CFU/mL).

### 2.2. Preparation of the Dental Root Canals

A total of 50 extracted non-decayed human wisdom teeth with round and straight roots were collected and stored in normal saline at 4 °C until use. The study was approved by the Ethics Committee of the Faculty of Medicine, University Hospital Jena, Germany (#2019-1401-Material). Informed consent of each patient was given.

Prior to any treatment, soft tissue remands were removed from the external root surfaces with curettes (Hu-Friedy, Frankfurt am Main, Germany). Subsequently, the crowns were removed using a water-cooled cut-off wheel (Horico Dental, Berlin Germany) to receive final root lengths of 10 mm. The pulp tissue was removed and the root canals were manually enlarged with files (Densply Detrey GmbH, Konstanz, Germany) up to size ISO 50. For irrigation normal saline was used. To ensure a leak-proof outer insolation, the root surfaces were coated with glass ionomer cement (Ketac Bond, 3M ESPE, Neuss, Germany). For more convenient handling, the prepared root specimens were embedded in Flexitime impression putty material (Heraeus Kulzer GmbH, Hanau, Germany). In Figure 1 the enlargement of the main root canal with endodontic files and irrigation with normal saline is shown.

### 2.3. Inoculation of the Root Canals with E. faecalis

Prior to inoculation, all canals were irrigated with 2 mL Calcinase (lege artis Pharma GmbH & Co KG, Dettenhausen, Germany) for 180 s and afterwards rinsed with 3 mL distilled water each. Subsequently, all specimens were autoclaved in a moist chamber for 20 min at 121 °C.

After sterilization, all root canals were inoculated with 100 μL of the E. faecalis suspension (OD_546nm_ of 0.5) for 48 h under anaerobic standard conditions (80% N_2_, 10% CO_2_, and 10% H_2_). To avoid drying of the samples, all specimens were moistened with Schaedler fluid media after 24 h of incubation.

### 2.4. Microscopic Evaluation of the Infected Root Specimens

After 48 h bacterial colonization of the dentinal tubules was observed by scanning electron microscopy. Therefore, the infected specimens were prepared as described at Kneist et al. [29]. In brief, after infection with *E. faecalis* the roots were fractured and fixed with 2.5% glutaraldehyde/cacodylate buffer for 24 h, washed three times in cacodylate buffer and subsequently dehydrated for 30–60 min by sequential washes in 30% to 100% graded ethanol. After critical-point drying, the specimens were sputtered with gold and observed using a scanning electron microscope (LEO 1450VP, Zeiss, Oberkochen, Germany) at 20–20,000× magnification.

In addition, remands of the fractured and fixed specimens were subjected to histological evaluation, too. Therefore, the fragments were additionally decalcified in 10% EDTA for 12 weeks at 37 °C, embedded into paraffin and cut by a microtome. Subsequently, all samples were stained according to Brown and Brenn. Evaluation was performed using brightfield microscopy at 40× magnification.

### 2.5. Root Canal Disinfection with Chloramine-T

All infected specimens (n = 50) were assigned into 5 randomized groups each with n = 10 samples. In test group 1 (chloramine-T CP), root canals were completely packed each with 7 mg of crystalline chloramine-T (concentration 99%) (Merck Chemicals GmbH, Darmstadt, Germany). In test group 2 (chloramine-T PP), sterile paper points were moistened with normal saline and subsequently dip coated with an average of 1.3 mg crystalline chloramine-T. Afterwards, the coated paper points were inserted into the infected root canals. Subsequently, all specimens were incubated in a moistened chamber for 24 h under anaerobic standard conditions (80% N_2_, 10% CO_2_, and 10% H_2_). Additionally, all canals were closed by a top seal with Flexitime impression putty material.

In control group 1 calcium hydroxide paste (Calcicur^®^, VOCO GmbH, Cuxhaven, Germany) was applied as intracanal dressing for 24 h. To simulate a saliva-proof seal, all canal orifices were closed with Flexitime impression putty material, too.

In test group 3 the root canals were irrigated for 20 s with 5 mL of a 1.3% sodium hypochlorite solution (Hedinger GmbH, Stuttgart, Germany). In control group 2 normal saline (5 mL) was used for irrigation.

### 2.6. Microbial Analysis

The applied methodology was adopted from Ossmann et al. [20]. In brief, after inoculation with *E. faecalis* for 48 h, the samples were dried twice for 10 s each with ISO 50 paper points (DentsplyDeTrey, Konstanz, Germany). The used paper points (ISO 50) served as initial colonization control (baseline value—P0) and were traversed to a reaction vessel filled with 1 mL of physiological NaCl solution. After vortexing, dilution series down to 10^−6^ were arranged and aliquots (100 µL) were plated onto sterile Schaedler agar. All plates were cultivated for 24 h under anaerobic standard conditions (80% N_2_, 10% CO_2_, and 10% H_2_) and colony-forming unites (CFU/mL) were determined.

After treatment with the respective disinfectant another swap (P50) was taken from the canal wall by inserting two consecutive sterile paper points that were analyzed in the same way as already described.

Subsequently, all canals were manually enlarged with endodontic hand-held files (ANTAEoS CC+, VDW GmbH, Munich, Germany) from size ISO 60 to size ISO 110 (P60 to P110). Each file was turned 10 times by three-quarter rotations. The abraded dentin debris of each shaping step was collected and transferred together with the used file to a reaction vessel filled with 1 mL of normal saline. The remaining dentin debris was taken from the root canal with another paper point and also transferred to the same reaction vessel. Shaping and collecting of the dentin debris was carried out twice for each ISO size.

Preparation from one to the other ISO size resulted in reduction of the dentin wall by app. 50 µm. Before proceeding with the shaping process, any residual dentin debris was removed from the root canals by rinsing each canal with 3 mL of normal saline. The rinsing fluid was discharged. Subsequently, the pooled samples were vortexed, dilution series down to 10^−6^ were arranged and aliquots (100 µL) were plated onto sterile Schaedler agar (Schaedler Anaerobic Agar, Oxoid Ltd., Hampshire, UK) supplemented with 6% sheep blood and 0.1% vitamin K (Konakion MM 10 mg, Roche Pharma AG, Basel, Switzerland). Afterwards, all plates were cultivated anaerobically for 48 h and CFU/mL were determined. The experimental setup is shown in Figure 2, too.

### 2.7. Data Analysis

Statistical analysis was carried out using SPSS 20.0 PC software (SPSS, Inc., Dallas TX, USA). Significant differences between the groups were determined by applying the students’ *t*-test. The level of statistical significance was *p* < 0.05. Error corrections according to Bonferroni were additionally applied.

## 3. Results

The present in vitro study investigated the antibacterial effect of temporary dressings with chloramine-T and calcium hydroxide as well as irrigations with 1.3% sodium hypochloride on *E. faecalis* inside infected root canals. Chloramine-T was applied to the infected samples by either complete packing of the main root canal with solid chloramine-T crystals (Chloramine-T CP) or by moistened and dip-coated paper points (chloramine-T PP). As control irrigation with normal saline was used.

Prior to any antibacterial testing, colonization of the root canal system by *E. faecalis* was evaluated by microscopic examination. In Figure 3a–c scanning electron microscopic images (SEM) of infected root dentin samples are shown. It can be observed that *E. faecalis* invaded the dentinal tubules up to distances of at least 300 µm. A magnified image of lined up bacterial cells inside a dentinal tubule is shown in Figure 3c.

Within 48 h of inoculation *E. faecalis* was also able to form biofilms at the walls of the main root canals. In Figure 3d bacterial cells adhering to the root canal wall and also cells that have invaded the dentinal tubules are shown. A detailed image of bacterial cells inside a dentinal tubule can also be viewed in Figure 3e.

In order to evaluate the efficiency of each antimicrobial measure, the main root canal was enlarged stepwise from ISO 50 to ISO 110 and the abraded dentin debris was subjected to microbial culture analysis. Prior to any antimicrobial testing an average of 1.58 × 10^7^ CFU/mL (P0 value—baseline) cells was estimated that adhered to the main root canal wall. In accordance to the microscopic evaluation, microbial culture analysis confirmed colonization of the dentinal tubules by *E. faecalis* up to distances of 300 µm, too (Figure 4, NaCl control, dashed line).

Among all disinfectants tested, complete filling of the main root canal with crystalline chloramine-T (chloramine-T CP) for 24 h was most efficient in suppressing *E. faecalis*. As shown in Figure 3 complete filling of the main root canal with chloramine-T resulted in total suppression of the species. Culture analysis of the collected dentin debris confirmed complete suppression of *E. faecalis* up to file size ISO 110, which corresponds with a distance of penetration into the dentinal tubules by up to 300 µm.

In comparison, when chloramine-T was applied by moistened and dip-coated paper points (chloramine-T PP—test group 2) only bacteria that adhered to the walls of the main root canal were affected (ISO size 50). Microbial culture analysis estimated a reduction in CFU/mL by 3 log-counts when compared to the P0 value. Bacteria that have invaded the dentinal tubules remained unaffected by this kind of disinfection measure. Vital cells could be recovered from the collected dentin debris in high numbers up to file size ISO 110.

In the following section results from the test and control groups are compared. There was no significant difference in the results obtained for the chloramine-T PP group when compared to the CaOH control. It was found that application of CaOH for 24 h resulted in a decrease by 3.9 log-counts of cells that colonized the canal walls directly, which was significantly not different from the results obtained for the chloramine-T PP group (*p* = 0.065). Further mechanical instrumentation of the main root canal up to file size ISO 110 did not reveal any significant change in the number of bacteria recovered when chloramine-PP was compared to the suppressive behavior of the CaOH control.

Treatment with CaOH resulted in significant bacterial reduction inside the dentinal tubules which ranged in between 3.0 (ISO 60) and 2.6 (ISO 110) log counts when compared to the P0 value. Compared to chloramine-T CP, temporary dressings with CaOH only showed a minor antibacterial effect on *E. faecalis* inside the dentinal tubules.

Further, it was found that irrigation with 1.3% NaOCl was efficient in suppressing *E. faecalis*, too. Treatment for 20 s resulted in complete suppression of bacteria at the root canal walls, but failed to suppress the species inside the dentinal tubules completely.

When compared to the NaCl-control, treatment with 1.3% NaOCl caused significant bacterial suppression up to a penetration depth of 250 µm (ISO 100, *p* = 0.001). The correlation between ISO size and depth of bacterial penetration (distance of bacterial invasion into the dentinal tubules) is illustrated in Figure 5.

## 4. Discussion

The aim of the present in vitro study was to investigate the antibacterial effect of N-chlorotosylamidesodium salt (chloramine-T), CaOH and NaOCl on *E. faecalis* in infected dentinal tubules. Crystalline chloramine-T was delivered to the main root canals by either complete packing or by moistened and dip-coated paper points.

In the present study colonization of the dentinal tubules by *E. faecalis* was verified by scanning electron microscopy up to distances of at least 300 μm. Besides SEM evaluation, bacterial colonization was also confirmed by microbiological cultures analysis. Therefore our group developed and adapted an examination model that was published recently [21]. The applied method is based upon a stepwise enlargement of the main root canal with hand-held endodontic files and delivers detailed information upon the distance of bacterial penetration and bacterial load inside the dentinal tubules.

In a recent study, our group was also able to examine the colonization behavior of *E. faecalis* by using artificial SiO/SiO_2_-micro-tubes. It was shown that bacterial colonization is strongly dependent upon the diameter and length of the tubes. Detailed examination revealed that within the same inoculation time tubes of wider diameter were colonized to a stronger extend compared to those of a smaller diameter. Further, a distance of penetration up to 500 µm was verified for *E. faecalis* [30].

However, colonization of dentinal tubules by *E. faecalis* has been studied by many authors so far [2,21,31,32]. It was found, that the species, unlike other microbes, is able to invade the dentinal tubules also by itself [33]. Once entered, *E. faecalis* can persists inside the tubules by adhering to collagen type-I fibers [2]. Because the species is well adapted to the predominant environment inside the dentinal tubules, colonization and biofilm formation begins quickly [34,35,36,37]. In this context, the present in vitro study confirmed that *E. faecalis* was able to colonize the dentinal tubules up to distances of at least 300 µm (ISO 110, Figure 3 and Figure 4).

From all antiseptics tested, chloramine-T presented the strongest antibacterial activity. Dressings for 24 h caused total suppression of bacteria that have invaded the dentinal tubules.

In solid state, chloramine-T is of crystalline appearance and formed by a reaction of chlorine with nitrogenous compounds. In aqueous surrounding chloramine-T dissociates into sodium hypochloride (Figure 6A) and further into active hypochloric acid (Figure 6B).

In contrast to NaOCl that is commonly used in chemo-mechanical root canal preparation, the release of reactive chlorine from chloramine-T occurs rather slowly which results in prolonged antibacterial activity [24,38]. Because of the dissociation kinetics, the overall antibacterial effect of chloramine-T is much stronger compared to irrigations with NaOCl.

During the dissociation process, microorganisms such as *E. faecalis* are lethally damaged by oxidative reactions [39]. Investigations have shown that contact of microorganisms to chloramine-T results in distraction of the cell wall and an increase in permeability [28]

Within the limitations of this study, it was shown that chloramine-T delivered by paper points showed an insufficient antibacterial effect. Due to the lack of preliminary studies, investigations that evaluate the optimum concentration of chloramine-T, especially for paper point application are still needed.

If higher drug concentrations were applied, antibacterial efficiency would have probably increased. However, stronger concentrations will cause an increase in cytotoxicity, too. Because the dental roots are surrounded by different soft and hard tissues, drugs that diffuse out of the root canal system might cause severe irritations, leading to persistent signs of inflammation.

In clinical practice NaOCl is commonly applied during chemo-mechanical root canal preparation in concentrations normally ranging in between 0.5 and 5%. NaOCl acts antimicrobial and also sows tissue dissolution abilities. When applied to root canals, immediate reactions with surrounding organic material will take place, leading to quick depletion. In clinical practice NaOCl solution is therefore commonly applied for irrigation purpose (short application time), only.

Guiteras et al. compared the activity of NaOCl and chloramine-T on organic substrates. It was found that in case of NaOCl available chlorine was detected for 1 h, only. In contrary, chloramine-T showed available chlorine for more than 24 h [40].

As shown in Figure 6, salvation of chloramine-T needs to take place first. Afterwards dissociation to active NaOCl occurs. In comparison to irrigation with NaOCl fluid, chloramine-T is therefore antibacterial for a much longer period of time.

Other than irrigation with NaOCl fluid, chloramine-T was applied in the present study by means of a temporary antiseptic root canal filling for 24 h. Because chloramine-T shows a prolonged antibacterial effect, suppression of bacteria, especially inside dentinal tubules, is much more efficient compared to irrigations with NaOCl.

In the present study, the effect of chloramine-T was also compared to the antibacterial effect of a temporary dressing with calcium hydroxide. It was found that crystalline chloramine-T was most efficient in suppressing *E. faecalis*, while application of calcium hydroxide only resulted in minor bacterial suppression inside the dentinal tubules (Figure 4). The results are in line with findings of other authors that reported upon a reduced antimicrobial activity of calcium hydroxide on *E. faecalis* in infected dental root canals, too [13,41].

However, calcium hydroxide is still one of the most recommended temporary dressing agent applied [42]. Calcium hydroxide also presents some antimicrobial activity that is mainly based upon the dissociation of the compound into calcium and hydroxyl ions which cause distraction of lipopolysaccharides inside the bacterial cell wall [43,44]. The dissociation thereby is also influenced by the kind of calcium hydroxide formulation applied and was found to be strongest for compositions with polyethylene glycol [45]. However, overall, the antibacterial activity of calcium hydroxide on *E. faecalis* is often rather insufficient [13,14,46].

In the present study the effect of irrigations with sodium hypochlorite solution on *E. faecalis* was also observed. So far, authors already confirmed the antibacterial activity of NaOCl on *E. faecalis* and showed that the effect is strongly dependent upon the applied concentration and irrigation time [47,48]. As shown in the present study, irrigation with 1.3% NaOCl solution caused complete suppression of bacteria that colonized the walls of the main root canals directly, but failed complete suppression inside the dentinal tubules. The applied concentration was rather low. Application of a stronger solution might probably also cause an increase in antibacterial efficiency. Additionally, ultrasonic agitation and the use of pulsed lasers for activation are common strategies in order to increase the disinfection capability, but were not applied in the present study neither [49,50].

The antibacterial effect of sodium hypochlorite, similar to calcium hydroxide, can thereby also be referred to its alkaline pH [51]. Contact of microorganisms to NaOCl causes peroxidation of lipids and leads to damage of cytoplasmic membranes. In addition, it was found that a reaction of NaOCl with surrounding organic material results in the formation of chloramines that themselves have a negative effect on the cells’ metabolism [51,52].

Recently, the effect of chloramine-T on *E. faecalis* was also observed by Wang et al. It was ascertained that chloramine-T shows promising results [53]. In accordance, a former microscopic based analysis highlighted the potential of chloramine-T as efficient disinfectant in the treatment of infected dental root canals, too [54].

The results of the present study demonstrated that from all disinfectants tested, crystalline chloramine-T showed the strongest antibacterial activity and was even capable in complete suppression of bacteria that have penetrated far into the dentinal tubules. In conclusion, it can be assumed that application of chloramine-T might even help to improve endodontic disinfection. As shown in a recent study, the maintenance of teeth by proper endodontic therapy should in the most cases be the first options in order to prevent insertion of cost intensive dental implants [55]. Further clinical investigations are necessary in order to proof the efficacy and safety of chloramine-T also in vivo.

## 5. Conclusions

The results of the present in vitro study proofed that from all antiseptics tested, chloramine-T showed the strongest antibacterial activity. Unlike treatment with NaOCl and CaOH, application of crystalline chloramine-T was also efficient in eradicating bacteria that have penetrated far into the dentinal tubules. In the case of chloramine-T, a disinfection range of 300 µm was estimated. In conclusion, chloramine-T shows promising characteristics and might be an auspicious alternative to antimicrobials commonly used in endodontic antiseptic therapy.

## Figures and Tables

**Figure 1 materials-14-02427-f001:**
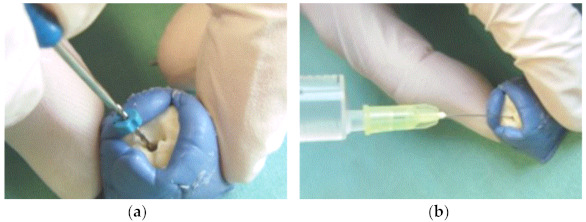
The picture shows root specimens embedded in putty material: (**a**) enlargement of the main root canal with endodontic files; (**b**) irrigation of the shaped canal with normal saline.

**Figure 2 materials-14-02427-f002:**
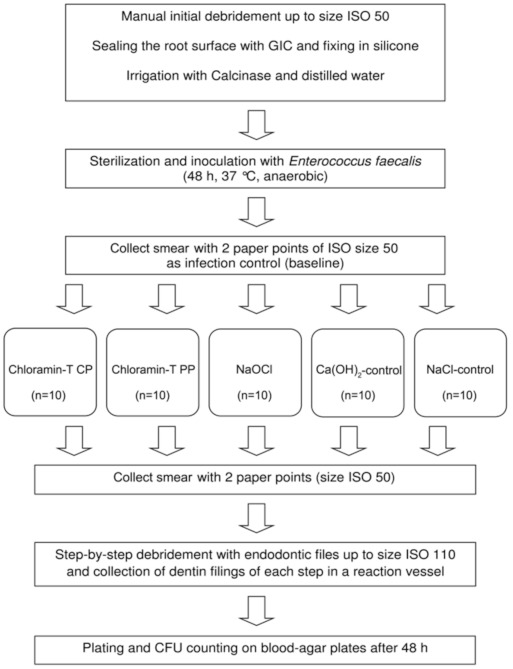
Flow-chart of the experimental setup. GIC-glass ionomere cement.

**Figure 3 materials-14-02427-f003:**
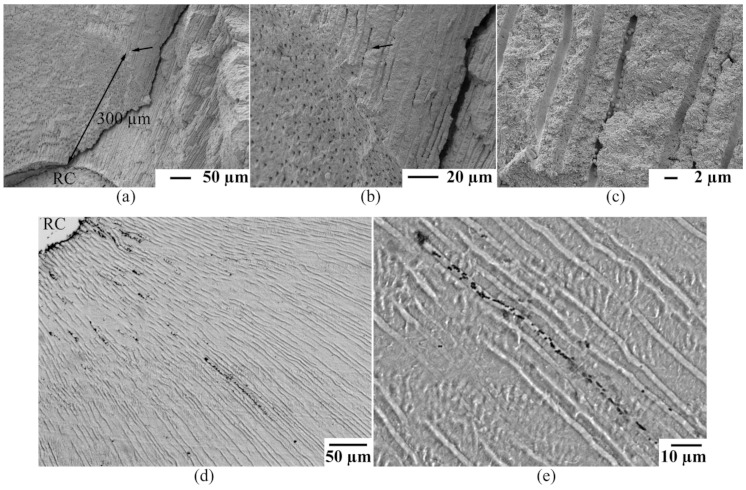
Scanning electron microscopic images (**a**–**c**) and histological sections (**d**,**e**) of root dentin samples showing bacteria that have invaded the dentinal tubules: (**a**–**c**) Series of magnified dentinal tubules. Bacteria inside a dentinal tubule are marked by a black arrow. The main root canal is labeled with RC; (**d**,**e**) Histological section and magnification of fixed bacterial cells inside dentinal tubules.

**Figure 4 materials-14-02427-f004:**
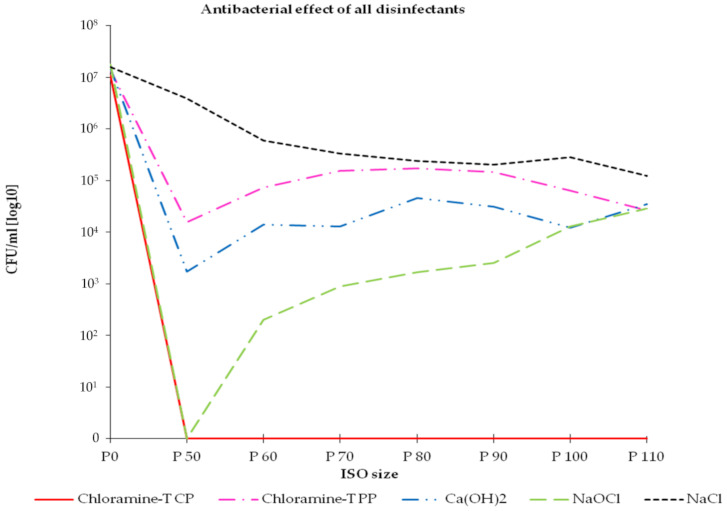
Antibacterial effect of all disinfectants that were applied. All root canals were manually enlarged from file size ISO 50 (P50) to size ISO 110 (P110). The collected dentin debris was subjected to microbial culture analysis. Microbial load of each ISO size is shown in CFU/mL. The baseline value (bacterial load of the main root canal before any antimicrobial measure) is termed P0. Additional graphical illustrations regarding the antimicrobial effect are attached as Appendix A.

**Figure 5 materials-14-02427-f005:**
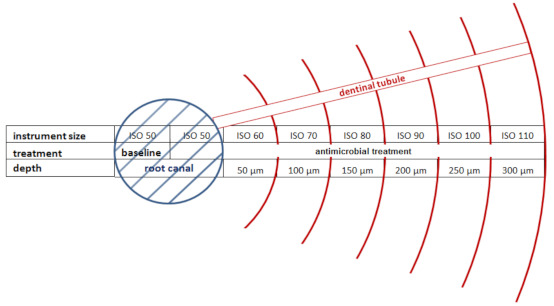
The image visualizes the correlation between ISO size of the applied endodontic file and depth of bacterial invasion into the dentinal tubules (image adopted and modified from Ossmann et al. [21]). Each manual instrumentation step with hand-held endodontic files resulted in an enlargement of the main root canal by 50 µm. Instrumentation up to file size ISO 110 resulted in abrasion of the root dentin by 300 µm.

**Figure 6 materials-14-02427-f006:**
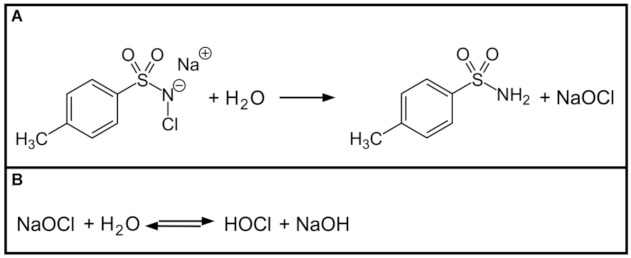
Dissociation of chloramine-T and NaOCl. (**A**) In the presence of water chloramine-T dissociates into NaOCl. (**B**) As in case of sodium hypochlorite, the active agent is hypochloric acid.

## Data Availability

Not applicable.

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
