# Peer review of "Antibacterial Effect of Endodontic Disinfections on Enterococcus Faecalis in Dental Root Canals—An In-Vitro Model Study"

_materials, 2021, doi:10.3390/ma14092427_

Round 1

Reviewer 1 Report

The authors investigated different disinfection methods for E. faecalis cultured on root specimens. This paper includes a feasible application of different disinfection methods but the experimental design is not technically sound.

The major concerns of this study:

  1. A critical point of this study is that the authors used different disinfection methods; wondering if different concentrations of Chloramine-T with/without paper points give similar effect. How the author compare disinfection methods in a different procedure of treatment (i.e., different exposure time)? Did authors use standardized procedure of each method? What happen if time is changed during the treatment of NaOCl (20s vs 24 h)? I understand it might be clinically relevant but wondering time and concentration used in chloramine-T TC is equivalent to NaOCl used in this study. What happen if NaOCl generated in a certain concentration of chloramine-T and an equivalent concentration of NaOCl is test for 24 h?
  2. In Microscopic evaluation, did authors use confocal microscopy with fluorescent dye labeling? Or just used bright field for observation of sectioned specimens? Please clarify the methods.
  3. In Figure 3, if the line indicates mean, please provide SD bar.
  4. In Figure 5, it is not including ‘kinetics’. Where is the time (rate)? It seems to be scheme for dissociation of chloramine-T and NaOCl.

Minor:

  1. Please describe the details on ‘anaerobic standard condition’
  2. Figure 1, sequence of treatment group should be consistent with methodology and data presentation.
  3. Line 166, antibacterial effect would be appropriated because the authors did not assess the bactericidal effect of agents in this study.

Author Response

Dear Reviewer,

we say thank you for all comments that helped us to improve the scientific appearance of our study. Please find below the answers to your comments. All changes within the publication are highlighted in yellow. We tried to answer each question to the best of our knowledge. Once again, we are very thankful for you reviewing our manuscript.

The authors investigated different disinfection methods for E. faecalis cultured on root specimens. This paper includes a feasible application of different disinfection methods but the experimental design is not technically sound.

The major concerns of this study:

Question: A critical point of this study is that the authors used different disinfection methods; wondering if different concentrations of Chloramine-T with/without paper points give similar effect.

Answer: The question is aimed on the activity of different Chloramine-T concentrations. Stronger concentrations will probably raise the antibacterial efficiency, but will also induce a stronger cytotoxic effect. Because the dental roots are surrounded by different soft and hard tissues, drugs that diffuse out of the root canal system might cause severe irritations, leading to persistent signs of inflammation. In the present study Chloramine-T was applied to the main root canals by means of a temporary dressing for 24h. Since the applied concentration of Chloramine-T is efficient in eliminating all bacteria colonizing the root canal system, we decided not to apply stronger drug concentrations. As shown, Chloramine-T applied by paper points caused an insufficient antibacterial effect. Therefore, further reduction in Chloramine-T will cause less antibacterial behavior.      

Question: How the author compare disinfection methods in a different procedure of treatment (i.e., different exposure time)?

Answer: The reviewer is right by criticizing the mode of comparison. We therefore corrected parts of the results and materials section. Now, the results are compared according to the treatment mode (irrigation/ temporary dressing). In detail, Chloramine-T CP and Chloramine-T PP are now compared to temporary dressings with CaOH paste (CaOH-control) and irrigations with NaOCl are compared to treatment with the NaCl-control. This implicated corrections among figure 2, too.

Question: Did authors use standardized procedure of each method? What happen if time is changed during the treatment of NaOCl (20s vs 24 h)?

Answer: Each method was applied in a standardized manner. All treatment procedures are adopted from clinical protocols. The evaluation method applied was published prior. For further information please view: Ossmann, A.; Kranz, S.; Andre, G.; Volpel, A.; Albrecht, V.; Fahr, A.; Sigusch, B. W., Photodynamic killing of Enterococcus faecalis in dentinal tubules using mTHPC incorporated in liposomes and invasomes. Clinical oral investigations 2015, 19, (2), 373-84.

If NaOCl is applied in higher concentrations, antibacterial efficiency will certainly increase. NaOCl is commonly applied as irrigation fluid during chemo-mechanical root canal preparation. NaOCl acts antimicrobial and solves remaining pulp tissue. When applied to the root canals the compound instantly reacts with surrounding organic material which causes quick depletion. In clinical practice irrigations with NaOCl are therefore applied for short periods of time (seconds to minutes), only. Because the antimicrobial activity of NaOCl is of short duration, incubation for 24h will not bring any benefit.

In case of Chloramine-T, which is of solid appearance, salvation will take place first. Afterwards dissociation to active NaOCl occurs. Therefore Chloramine-T acts antibacterial for a longer period of time. Other than irrigation with NaOCl fluid, Chloramine-T was applied by means of a temporary antiseptic root canal filling for 24h. Because Chloramine-T shows a prolonged antibacterial effect, suppression of bacteria inside dentinal tubules is much more efficient compared to irrigation with NaOCl fluid, only.    

Question: I understand it might be clinically relevant but wondering time and concentration used in chloramine-T TC is equivalent to NaOCl used in this study. What happen if NaOCl generated in a certain concentration of chloramine-T and an equivalent concentration of NaOCl is tested for 24 h?

Answer: The reviewer is right. As already mentioned above, we now compare Chloramine-T with temporary dressings of CaOH. Irrigation with NaOCl is now compared to irrigation with normal saline. We are very sorry for this mistake! Further, the concentration of NaOCl and Chloramine-T is not equivalent. In clinical practice concentrations of NaOCl ranging in between 0.5 and 5% are applied. This study uses 1.3%. An increase in concentration > 5% will cause damage to the alveolar tissue surrounding the root. In addition, NaOCl will not show significant antibacterial action after incubation with organic material for 24h anymore. As mentioned in the conclusions, too, further studies are necessary in order to observe the safety and efficiency of Chloramin-T also in-vivo.

Question: In Microscopic evaluation, did authors use confocal microscopy with fluorescent dye labeling? Or just used bright field for observation of sectioned specimens? Please clarify the methods.

Answer: Thank you for this remark. The specimens were observed by brightfield microscopy. We corrected the mistake within the publication.

Question: In Figure 3, if the line indicates mean, please provide SD bar.

Answer: We regret from providing SD bars because they will crowd the diagram. We attached supplementary figures (Figure S1-S4) showing the results in boxplots.

Question: In Figure 5, it is not including ‘kinetics’. Where is the time (rate)? It seems to be scheme for dissociation of chloramine-T and NaOCl.

Answer: Thank you for this remark. We have deleted the term ‘kinetics’

Question: Please describe the details on ‘anaerobic standard condition’

Answer: We have added information defining the anaerobic standard conditions. For cultivation an anaerobic atmosphere was composed of 80% N2, 10% CO2, and 10% H2 in a closed incubator.

Question: Figure 1, sequence of treatment group should be consistent with methodology and data presentation.

Answer: We corrected the sequence of presentation among the results section. Thank you for this comment!

Question: Line 166, antibacterial effect would be appropriated because the authors did not assess the bactericidal effect of agents in this study.

Answer: We changed ‘bactericidal’ to ‘antibacterial’

Reviewer 2 Report

Dear authors, thanks to provide this research. I suggest some minor revision to improve the quality of the manuscript. Moreover, please add some pictures to the samples preparations.

In addition, I would like to read a short statement in the discussion and/or introduction regarding the importance to maintain teeth. This research may also improve the efficacy of endodontic treatment or re-treatment, avoiding to much dental implants. You can refer to "1.Esposito, M., Sbricoli, L., Buti, J., Uccioli, U. & Tallarico, M. Endodontic retreatment of teeth with uncertain endodontic prognosis versus dental implants: 5-year results from a randomised controlled trial. Clinical Trials in Dentistry2, 27–43 (2020)."

Author Response

Dear Reviewer,

we say thank you for all comments that helped us to improve the scientific appearance of our study. Please find below the answers to your comments. All changes within the publication are highlighted in yellow. We tried to answer each question to the best of our knowledge. Once again, we are very thankful for you reviewing our manuscript.

Question: I suggest some minor revision to improve the quality of the manuscript. Moreover, please add some pictures to the samples preparations.

Answer: Pictures showing the root specimens embedded in putty material were introduced to the publication. In picture 1a the mechanical enlargement of the main root canal by hand-held endodontic files is shown. In picture 1b irrigation of the main root canal with 1.3% NaOCl is displayed.

Question: In addition, I would like to read a short statement in the discussion and/or introduction regarding the importance to maintain teeth. This research may also improve the efficacy of endodontic treatment or re-treatment, avoiding to much dental implants. You can refer to "1.Esposito, M., Sbricoli, L., Buti, J., Uccioli, U. & Tallarico, M. Endodontic retreatment of teeth with uncertain endodontic prognosis versus dental implants: 5-year results from a randomised controlled trial. Clinical Trials in Dentistry2, 27–43 (2020)."

Answer: We incorporated information upon the issue to the discussion. For more detailed information please view page 9 of the discussion. We included: Esposito, M.; Trullenque-Eriksson, A.; Tallarico, M., Endodontic retreatment versus dental implants of teeth with an uncertain endodontic prognosis: 3-year results from a randomised controlled trial. European journal of oral implantology 2018, 11, (4), 423-438 

Round 2

Reviewer 1 Report

Question: A critical point of this study is that the authors used different disinfection methods; wondering if different concentrations of Chloramine-T with/without paper points give similar effect.

Answer: The question is aimed on the activity of different Chloramine-T concentrations. Stronger concentrations will probably raise the antibacterial efficiency, but will also induce a stronger cytotoxic effect. Because the dental roots are surrounded by different soft and hard tissues, drugs that diffuse out of the root canal system might cause severe irritations, leading to persistent signs of inflammation. In the present study Chloramine-T was applied to the main root canals by means of a temporary dressing for 24h. Since the applied concentration of Chloramine-T is efficient in eliminating all bacteria colonizing the root canal system, we decided not to apply stronger drug concentrations. As shown, Chloramine-T applied by paper points caused an insufficient antibacterial effect. Therefore, further reduction in Chloramine-T will cause less antibacterial behavior.

>>Following the authors response, I have a further question that the authors need to address. Due to the lack of preliminary study evaluating the optimum concentration of Chloramine-T for paper point application, still this study is incomplete. The authors a least should discuss this point in the manuscript. Neither Chloramine-T CP nor chlormaine-T PP is entirely right; a high concentration used in Chloramine-T may have a side effect (cytotoxicity)/ lower concentration used in paper point is not sufficient for disinfection methods.

Question: Did authors use standardized procedure of each method? What happen if time is changed during the treatment of NaOCl (20s vs 24 h)?

Answer: Each method was applied in a standardized manner. All treatment procedures are adopted from clinical protocols. The evaluation method applied was published prior. For further information please view: Ossmann, A.; Kranz, S.; Andre, G.; Volpel, A.; Albrecht, V.; Fahr, A.; Sigusch, B. W., Photodynamic killing of Enterococcus faecalis in dentinal tubules using mTHPC incorporated in liposomes and invasomes. Clinical oral investigations 2015, 19, (2), 373-84.

If NaOCl is applied in higher concentrations, antibacterial efficiency will certainly increase. NaOCl is commonly applied as irrigation fluid during chemo-mechanical root canal preparation. NaOCl acts antimicrobial and solves remaining pulp tissue. When applied to the root canals the compound instantly reacts with surrounding organic material which causes quick depletion. In clinical practice irrigations with NaOCl are therefore applied for short periods of time (seconds to minutes), only. Because the antimicrobial activity of NaOCl is of short duration, incubation for 24h will not bring any benefit.

In case of Chloramine-T, which is of solid appearance, salvation will take place first. Afterwards dissociation to active NaOCl occurs. Therefore Chloramine-T acts antibacterial for a longer period of time. Other than irrigation with NaOCl fluid, Chloramine-T was applied by means of a temporary antiseptic root canal filling for 24h. Because Chloramine-T shows a prolonged antibacterial effect, suppression of bacteria inside dentinal tubules is much more efficient compared to irrigation with NaOCl fluid, only.

>>Some of points would be helpful for the reader; This is a broad-scope journal, please incorporate the information as described above “why a short period of time in NaOCl irrigations/root canal filling of Chloramine-T for 24 h is used in the clinical practice.”

Question: I understand it might be clinically relevant but wondering time and concentration used in chloramine-T TC is equivalent to NaOCl used in this study. What happen if NaOCl generated in a certain concentration of chloramine-T and an equivalent concentration of NaOCl is tested for 24 h?

Answer: The reviewer is right. As already mentioned above, we now compare Chloramine-T with temporary dressings of CaOH. Irrigation with NaOCl is now compared to irrigation with normal saline. We are very sorry for this mistake! Further, the concentration of NaOCl and Chloramine-T is not equivalent. In clinical practice concentrations of NaOCl ranging in between 0.5 and 5% are applied. This study uses 1.3%. An increase in concentration > 5% will cause damage to the alveolar tissue surrounding the root. In addition, NaOCl will not show significant antibacterial action after incubation with organic material for 24h anymore. As mentioned in the conclusions, too, further studies are necessary in order to observe the safety and efficiency of Chloramin-T also in-vivo.

>>As mentioned above, it would be great if the authors discuss this point in the manuscript.

Author Response

Dear Reviewer,

we say thank you for all further statements. Please find below the answers to your comments. All changes within the publication are highlighted in green. We tried to answer each question to the best of our knowledge and hope that our submission is now scientific sound. Once again, we are very thankful for you reviewing our manuscript.

Question R1: A critical point of this study is that the authors used different disinfection methods; wondering if different concentrations of Chloramine-T with/without paper points give similar effect.

Answer R1: The question is aimed on the activity of different Chloramine-T concentrations. Stronger concentrations will probably raise the antibacterial efficiency, but will also induce a stronger cytotoxic effect. Because the dental roots are surrounded by different soft and hard tissues, drugs that diffuse out of the root canal system might cause severe irritations, leading to persistent signs of inflammation. In the present study Chloramine-T was applied to the main root canals by means of a temporary dressing for 24h. Since the applied concentration of Chloramine-T is efficient in eliminating all bacteria colonizing the root canal system, we decided not to apply stronger drug concentrations. As shown, Chloramine-T applied by paper points caused an insufficient antibacterial effect. Therefore, further reduction in Chloramine-T will cause less antibacterial behavior.

Question R2: Following the authors response, I have a further question that the authors need to address. Due to the lack of preliminary study evaluating the optimum concentration of Chloramine-T for paper point application, still this study is incomplete. The authors a least should discuss this point in the manuscript. Neither Chloramine-T CP nor chlormaine-T PP is entirely right; a high concentration used in Chloramine-T may have a side effect (cytotoxicity)/ lower concentration used in paper point is not sufficient for disinfection methods.

Answer R2: The reviewer is right. We therefore addressed this issue and included further information to the discussion. We also included reference # 40.

Question R1: Did authors use standardized procedure of each method? What happen if time is changed during the treatment of NaOCl (20s vs 24 h)?

Answer R1: Each method was applied in a standardized manner. All treatment procedures are adopted from clinical protocols. The evaluation method applied was published prior. For further information please view: Ossmann, A.; Kranz, S.; Andre, G.; Volpel, A.; Albrecht, V.; Fahr, A.; Sigusch, B. W., Photodynamic killing of Enterococcus faecalis in dentinal tubules using mTHPC incorporated in liposomes and invasomes. Clinical oral investigations 2015, 19, (2), 373-84.

If NaOCl is applied in higher concentrations, antibacterial efficiency will certainly increase. NaOCl is commonly applied as irrigation fluid during chemo-mechanical root canal preparation. NaOCl acts antimicrobial and solves remaining pulp tissue. When applied to the root canals the compound instantly reacts with surrounding organic material which causes quick depletion. In clinical practice irrigations with NaOCl are therefore applied for short periods of time (seconds to minutes), only. Because the antimicrobial activity of NaOCl is of short duration, incubation for 24h will not bring any benefit.

In case of Chloramine-T, which is of solid appearance, salvation will take place first. Afterwards dissociation to active NaOCl occurs. Therefore Chloramine-T acts antibacterial for a longer period of time. Other than irrigation with NaOCl fluid, Chloramine-T was applied by means of a temporary antiseptic root canal filling for 24h. Because Chloramine-T shows a prolonged antibacterial effect, suppression of bacteria inside dentinal tubules is much more efficient compared to irrigation with NaOCl fluid, only.

Question R2: Some of points would be helpful for the reader; This is a broad-scope journal, please incorporate the information as described above “why a short period of time in NaOCl irrigations/root canal filling of Chloramine-T for 24 h is used in the clinical practice.”

Answer R2: We included some of the information to the study.

Question R1: I understand it might be clinically relevant but wondering time and concentration used in chloramine-T TC is equivalent to NaOCl used in this study. What happen if NaOCl generated in a certain concentration of chloramine-T and an equivalent concentration of NaOCl is tested for 24 h?

Answer R1: The reviewer is right. As already mentioned above, we now compare Chloramine-T with temporary dressings of CaOH. Irrigation with NaOCl is now compared to irrigation with normal saline. We are very sorry for this mistake! Further, the concentration of NaOCl and Chloramine-T is not equivalent. In clinical practice concentrations of NaOCl ranging in between 0.5 and 5% are applied. This study uses 1.3%. An increase in concentration > 5% will cause damage to the alveolar tissue surrounding the root. In addition, NaOCl will not show significant antibacterial action after incubation with organic material for 24h anymore. As mentioned in the conclusions, too, further studies are necessary in order to observe the safety and efficiency of Chloramin-T also in-vivo.

Question R2: As mentioned above, it would be great if the authors discuss this point in the manuscript.

Answer R2: Thank you for this comment. Some of the information is now addressed in the discussion.